# A FULLY AUTOMATED PERIODICITY DETECTION IN TIME SERIES

## ABSTRACT

This paper presents a method to autonomously find periodicities in a signal. It is based on the same idea of using Fourier Transform and autocorrelation function presented in Vlachos et al. (2005). While showing interesting results this method does not perform well on noisy signals or signals with multiple periodicities. Thus, our method adds several new extra steps (hints clustering, filtering and detrending) to fix these issues. Experimental results show that the proposed method outperforms state of the art algorithms.

## INTRODUCTION

A time series is defined by its 3 main components : the trend component, the periodic component and the random component. Trend analysis and prediction are topics that have been greatly studied Saad et al. (1998) and will not be treated in the article, therefore every time series will be assumed stationary regarding its mean and variance, so this study focus the periodic component. The ability to detect and find the main characteristic of this component is not as easy as the trend component. Yet, the ability to detect periodicities in a time series is essential to make precise forecasts.

A periodicity is a pattern in a time series that occurs at regular time intervals. More precisely, the time series is said cyclical, if the time intervals at which the pattern repeats itself can't be precisely defined and is not constant. On the opposite, there are seasonal time series in which the pattern repeats itself at constant and well defined time intervals. Thus, cyclical patterns are more difficult to detect due to their inconsistency and the fact that they usually repeat themselves over large periods of time and therefore require more data to be identified. Nevertheless, seasonal patterns are very common in time series such as those related to human behaviour which usually have periodicities like hours and calendar (time of day, day of week, month of year). This kind of feature is well known and can be easily tested to see if they are beneficial or not. Unfortunately, when it comes to time series related to other phenomenons, the periodicities are not trivially found. For instance, tides level are multi-periodic time series correlated to both moon cycles and sun cycles; and females menstrual cycles are related to hormonal changes. The ability to detect periodicity in time series is fundamental when it comes to forecasting Koopman & Ooms (2006). Once a periodic pattern has been detected, numerous techniques can be used to model this later and improve forecasts Gooijer & Hyndman (2006). However, periodicities detection is not easy and has been greatly studied in the existing literature, but most of current techniques are unable to detect periodicities without the need of preprocessing data Yuan et al. (2017) or have trouble detecting multiple periodicities Vlachos et al. (2005). This paper is organised as follow: we first present the Fourier transform and the Autoperiod algorithm Vlachos et al. (2005) used to detect periodicities in a signal. Then we propose a new fully automated method, named Clustered Filtered Detrended Autoperiod (CFD-Autoperiod), which also combines the advantages of frequency domain and time domain while being robust to noise and able to handle multi periodicities. Noise robustness is achieved using a density clustering on hints provided by the frequency analysis. Multi-periodicities are more precisely detected by both using detrending and filtering. Finally, we demonstrate that CFD-Autoperiod outperforms previous methods.

## RELATED WORKS

Autocorrelation and Fourier transform are well known techniques used to find recurrent patterns in a given signal.

### FOURIER TRANSFORM

The Fourier transform decomposes the original signal $\{s(t_j)\}_{j \in [1,N]}$ in a linear combination of complex sinusoids, also called a Fourier series. Let $N$ be the number of frequency components of a signal, $P$ the periodicity of the signal and $c_k$ the $k^{th}$ series coefficient then we have the Fourier series:

$$s_N(t) = \sum_{k=0}^{N-1} c_k \cdot e^{i\frac{2\pi kt}{N}}$$

Thus, the amplitude and phase of each sinusoids correspond to the main frequencies contained within the signal. The Fourier transform can easily detect the presence of frequencies in a signal. However, if we need the corresponding periodicities from each frequency then we have to return from the frequency domain to the time domain. Let $DFT$ be the Discrete Fourier Transform of a discrete signal $\{s(t_j)\}$, then we can obtain the corresponding Periodogram $\mathcal{P}$ in the time domain as follow:

$$\mathcal{P}(f_k) = ||DFT(f_k)||^2 = ||c_k||^2 \text{ with } k = 0, 1, ..., \lceil \tfrac{N-1}{2} \rceil$$

where $f_k = \frac{2\pi k}{N}$ correspond to the frequency captured by each component.

However, in the frequency domain each bin is separated with a constant step of $\frac{1}{N}$, whereas in the time domain bins size is $\frac{N}{k(k+1)}$, thus the range of periods is increasingly wider. Therefore, the Fourier transform resolution for long periods is not sufficient to provide an accurate estimation of the periodicity.

### AUTOCORRELATION

Another way to find the dominant periodicities in a signal consists in calculating the autocorrelation function (ACF) of the given signal $s(t)$. The autocorrelation is the correlation between the elements of a series and others from the same series separated from them by a given interval $\Delta t$:

$$ACF(\Delta t) = \frac{1}{N} \sum_{j=0}^{N-t} s(t_j) \cdot s(t_j + \Delta t)$$

The ACF function provides a more accurate estimation of each periodicity, especially for longer periods as opposed to the Fourier transform Vlachos et al. (2005). However, it is not sufficient by itself due to the difficulty to select the most predominant peaks. Indeed, for a given periodicity $p_1$ the autocorrelation generates peaks for each $p_1$ multiple, hence the difficulty to select the relevant peaks when multiple periodicities composed a signal.

### HYBRID APPROACH

A methodology combining both techniques advantages has been introduced by Vlachos et al. (2005). This method uses sequentially frequency domain (DFT) and time domain (ACF) in order to detect periodicity. The idea is to combine both methods in such a way that they complement each other. On the one hand, as mentioned earlier, due to its step inconstancy in the time domain, the Fourier transform resolution becomes insufficient to provide good estimations for long periods, where the autocorrelation has a constant resolution. On the other hand, according to Vlachos et al. (2005), it is difficult to correctly detect periodicities using only the autocorrelation function.

Thus they proposed the following steps: first, noise is discarded from possible periodicity hints using a threshold on the Periodogram. Then, these hints are refined using the ACF function. If a periodicity hint lies on a local maximum then it can be validated, otherwise, if it lies on a local minimum this latter is discarded. On top of that, thanks to the ACF resolution, a gradient ascent is used to refine the remaining hints Figure 1.

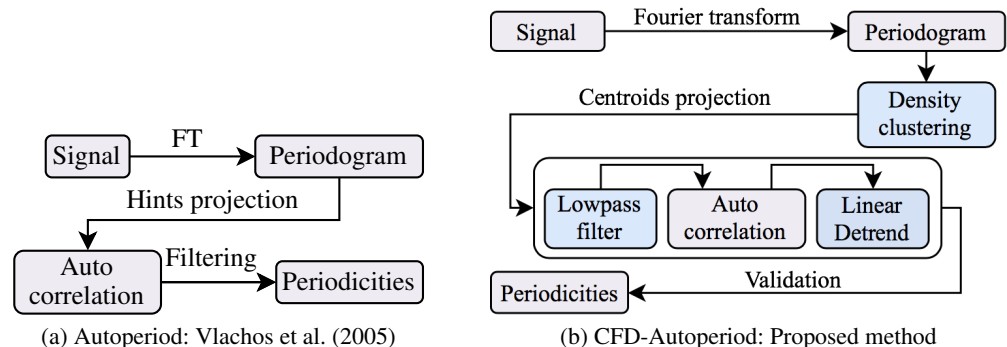

Figure 1: Periodicity detection methods

However, some issues such as multi-periodic signals, spectral leakage or presence of non-stationary periodicities are not addressed by the authors.

## A NEW METHODOLOGY: CFD-AUTOPERIOD

### SPECTRAL LEAKAGE

The Fourier Transform is used to select periodicity hints. To do so, we use the 99% confidence Li et al. (2010) Li et al. Vlachos et al. (2005) technique to compute the threshold distinguishing periodicity hints from noise in the Fourier transform. Firstly, it is necessary to find the maximum amount of spectral power generated by the signal noise. Let be $\{s'(t_j)\}_{j\in[1,N]}$ a permuted sequence of a periodic sequence $\{s(t_j)\}_{j\in[1,N]}$. $s'$ should not exhibit any periodic pattern due to the random permutation process. Therefore, the maximal spectral power generated by $s'$ should not be higher than the spectral power generated by a true periodicity in $s$. Thus, we can use this value as a threshold to eliminate the noise. To provide a 99% confidence level, this process is repeated 100 times and the $99th$ largest value recorded is used as a threshold.

Unfortunately, for a given periodicity in $X$, rather than finding an unique corresponding hint, spectral leakage may produce multiple hints near the true periodicity. This phenomenon is due to the finite resolution of the Fourier Transform and can only be avoided knowing in advance the true periodicities of the signal.

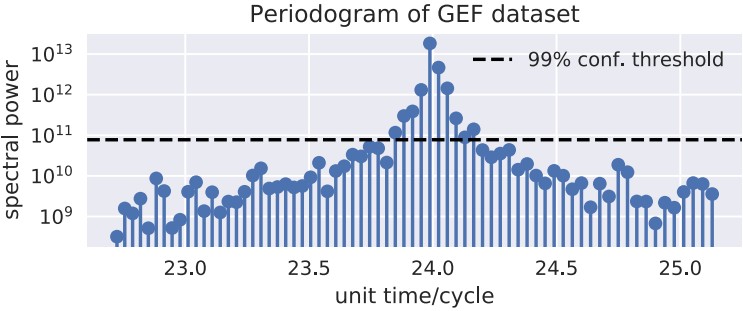

Figure 2: Illustration of spectral leakage

Spectral leakage generates points with a spectral power higher than the threshold provided by the 99% confidence method (Figure 2) and therefore generate imprecise periodicity hints. The autocorrelation might filter most of them but every imprecise periodicity hint increase the probability of false positives, therefore it is interesting to reduce the number of periodicity hints in order to achieve a higher precision score.

DENSITY CLUSTERING

Knowing that the distribution of spectral leakage is more dense around the true periodicity, performing a density clustering over periodicity hints and using the resulting centroids as periodicity hints can reduce the number of hints. A fundamental value in density clustering algorithms is the range in which it seeks for neighbours named $\epsilon$. In our case, this value is not a constant because the accuracy of the hint is related to the resolution of the corresponding DFT bin size. A hint may have leaked from adjacent DFT bins, thus for a given hint of periodicity $N/k$, $\epsilon$ is set as the next bin value plus a constant width of 1 to avoid numerical issues when the difference from the current bin value to the next bin value is less than one:

$$\varepsilon_{N/k} = \frac{N}{(k-1)} + 1$$

The clustering is done by ascending periodicity order, hence a cluster made with small periodicities cannot be altered by bigger periodicity clusters.

**Input:** Hints - list of hints in ascending order
**Output:** Centroids - list of centroids
1  Clusters ← [] ;
2  cluster ← [] ;
3  $\varepsilon$ ← Hints[0].nextBinValue + 1 ;
4  cluster.append(Hints[0]) ;
5  **for** *hint in Hints[1:]* **do**
6      **if** *hint $\leq \varepsilon$* **then**
7          cluster.append(hint) ;
8          $\varepsilon$ ← hint.binSize ;
9      **else**
10         Clusters.append(cluster) ;
11         cluster ← [] ;
12 **end**
13 Centroids ← [] ;
14 **for** *cluster in Clusters* **do**
15     centroid ← mean(cluster) ;
16     Centroids.append(centroid) ;
17 **end**

**Algorithm 1:** Clustering pseudocode

As shown in the results (Figure 3), the density clustering performed in the GEF dataset Hong et al. (2016b) drastically reduces the number of periodicity hints and the resulting centroids are close to the true periodicities ($24$ and $168$). Once the centroids have been found, they are used as periodicity hints during the validation step.

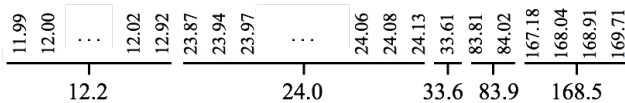

Figure 3: Density clustering results on GEF dataset. Top, before clustering and bottom, after.

HINTS VALIDATION

For the validation step, a search interval for each periodicity hint is needed to check if this latter lies on a hill or a valley of the ACF. Vlachos et al. (2005) used the DFT bin size to define this search interval but in this study we propose a different approach. A periodicity $N$ generates hills on the ACF at each multiple of $N$ and valleys at each multiple of $\frac{N}{2}$. Therefore, we defined the search interval $R$ for a periodicity hint $N$ as follow:

$$R = \left[\frac{N}{2}, ..., N + \frac{N}{2}\right]$$

Thereafter, a quadratic function is fitted to the ACR function in the search interval. In order to validate a hint, the function must have a negative second degree term and its derivative sign must change along the interval.

### MULTI-PERIODICITIES

The presence of multiple periodicities refutes the assumption that hills and valleys of the ACF are sufficient to validate or discard hints. Precisely, when validating a periodicity hint, correlations generated by both higher and lower frequencies than the hint can be problematic. These two problems are addressed in the following section.

### HIGHER FREQUENCIES

On the one hand, periodicities of higher frequencies induces sinusoidal correlations which may be in opposite phase with the correlation we are actually looking for (see Figure 4). Let $s$ be a multi-periodic signal composed of periodicities $P_1$ and $P_2$. Let $P_1$, a periodicity of length 20 and $P_2$, a periodicity of length 50. The periodicity $P_1$ produces on the signal ACF sinusoidal correlations of wavelength 20 and the periodicity $P_2$ produces sinusoidal correlations of wavelength 50. Thereby, at 50 lags on the ACF, the $P_1$ and $P_2$ periodicities will produce correlations in opposite phases and therefore nullify the hill at 50 used to validate or discard the periodicity hint $P_2$.

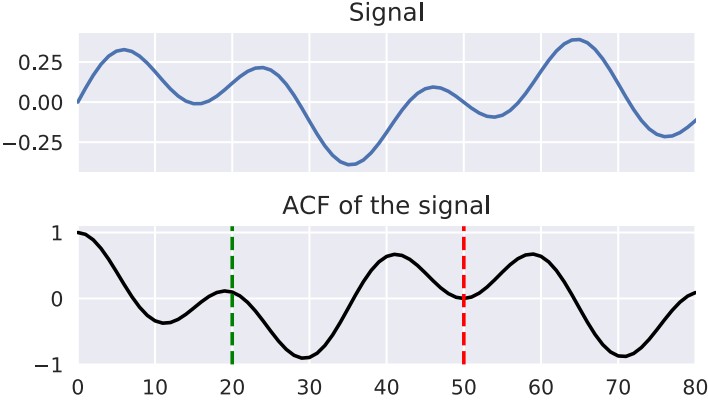

Figure 4: Impact of multiple periodicities (20 and 50) on the ACF.

To tackle this issue, periodicity hints are analysed in ascending order. If a periodicity hint is validated, a lowpass filter with an adapted cutoff frequency is applied to the signal. Consequently, the following autocorrelations will be computed on this new signal. Thus, the resulting autocorrelations will not exhibit any correlation induced by frequencies higher than the cutoff frequency of the lowpass filter.

The cutoff frequency must be chosen carefully. Indeed, an ideal lowpass filter is characterised by a full transmission in the pass band, a complete attenuation in the stop band and an instant transition between the two bands. However, in practice, filters are only an approximation of this ideal filter and the higher the order of the filter is, the more the filter approximates the ideal filter. In our case, we are studying the periodicities in the signal, therefore, we want a filter with a frequency response as flat as possible to avoid any negative impact on the periodicity detection. Thereby, a Butterworth filter has been chosen due to its flat frequency response with no ripples in the passband nor in the stopband.

However, a Butterworth filter, despite all the good properties, has a slow roll-off attenuating frequencies nearby the cutoff frequency. For the validation step, we do not want to attenuate the periodicity hint, therefore the cutoff frequency must not be exactly equal to the frequency of the hint. For a given periodicity $\frac{N}{k}$, the frequency cutoff is equal to the previous bin value minus 1, to avoid the

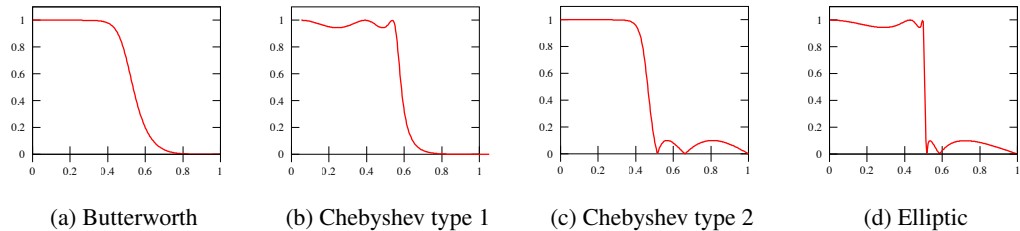

(a) Butterworth        (b) Chebyshev type 1        (c) Chebyshev type 2        (d) Elliptic

Figure 5: Illustration of differents lowpass filters.

same numerical issues as described in the Density Clustering section:

$$f_c = \frac{1}{(\frac{N}{k+1} - 1)}$$

LOWER FREQUENCIES

On the other hand, low frequencies may induce a local trend in the autocorrelation that can be problematic when validating an hint. Indeed, in order to validate a periodicity hint, a quadratic function is fitted to the ACF in the search interval as mentioned in the subsection Hints Validation. Sadly, a trend in the search interval may prevent the derivative sign to switch (Figure 6), and therefore prevent the correct validation of the corresponding hint.

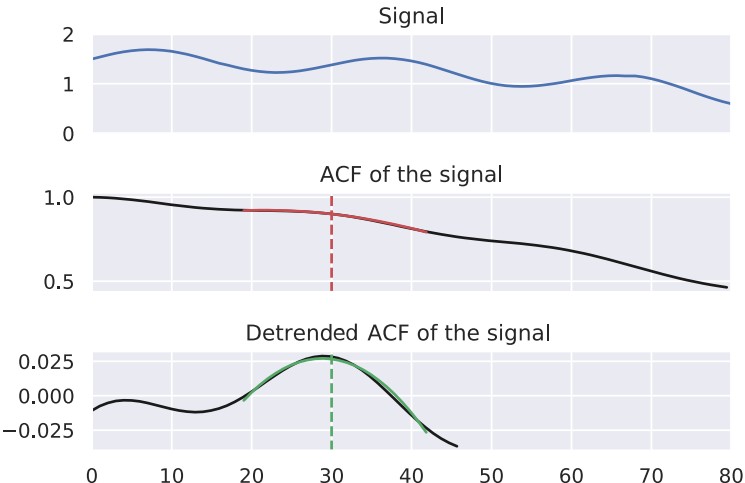

Figure 6: Hint validation using the ACF on a multi-periodic signal (30, 500).

Consequently, to avoid this situation, the ACF is detrended by subtracting the best fitted line in the following interval $[0, \frac{N}{(k-1)} + 1]$ for a given period hint $N/k$. Thus, the resulting ACF does not exhibit any linear trend and therefore the fitted quadratic function is able to validate or discard hint efficiently.

RESULTS

To evaluate the performances of the proposed method it is necessary to use time series datasets with periodicities. To do so, we perform our first evaluations on synthetic signals where the ground truth is known in order to compare raw performances and evaluations on real time series datasets.

SYNTHETIC SIGNALS

Signals of length 2000 with 1 to 3 periodicities have been generated. The periodicities have been chosen in the interval $[10, 500]$ using a pseudo-random process. For multi-periodic signals, this pseudo-random process ensures that periodicities are not overlapping each others by checking that one is at least twice as bigger as the previous one. Finally, in order to compute precision and recall metrics, a validation criterion has been established. We assumed that a periodicity $P_d$ detected in a generated signal with a true periodicity $P_t$ is valid if:

$$\lfloor 0.95 \times P_t \rfloor \leq P_d \leq \lceil 1.05 \times P_t \rceil$$

The metrics have been consolidated over 500 iterations for each generated periodicity. As shown in Table 1, for a non multi-periodic signal, autoperiod and CFD-Autoperiod method achieve high precision scores whereas the Fourier Transform achieves a high recall but a really low precision score. Indeed, the Fourier Transform method does not filter the hints using the autocorrelation. Nevertheless, the autoperiod method did not detect every periodicities even for non multi-periodic signals autoperiod. This is likely due to the absence of density clustering and the narrow interval search to find the corresponding hill on the ACF. For multi-periodic signals, both recall and precision are drastically decreasing for the autoperiod method and as it can be observed, the detrending step and the use of a lowpass filter by the CFD-Autoperiod method lead to better scores. Regarding the Fourier Transform scores, due to the lack of the filtering step its recall is high but its precision score is always the lowest.

| | | pseudo random | | | random | | |
|---|---|---|---|---|---|---|---|
| | Nb periods | 1 | 2 | 3 | 1 | 2 | 3 |
| Fourier transform | precision | 27.76 | 40.52 | 50.16 | 27.83 | 42.32 | 46.63 |
| | recall | 80.40 | 78.20 | **85.73** | 77.80 | **75.80** | **73.40** |
| autoperiod | precision | 98.47 | 64.16 | 54.42 | 98.39 | 58.05 | 53.41 |
| | recall | 77.20 | 51.20 | 32.87 | 73.20 | 35.70 | 26.13 |
| CFD-Autoperiod | precision | **100.00** | **91.10** | **86.93** | **100.0** | **71.78** | **68.07** |
| | recall | **100.00** | **91.10** | 78.93 | **100.0** | 55.20 | 43.07 |

Table 1: Precision/Recall comparison for pseudo-random and random process.

Benchmarks have also been performed on synthetic signals generated via random process, without limitations on the periodicity values (Table 1). Naturally, the results with an unique periodicity are similar. However, for multi-periodic signals the autoperiod and CFD-Autoperiod methods achieve lower scores. This is due to the fact that both methods use the autocorrelation to filter hints and this latter is not able to distinguish very close periodicities. Therefore, the use of autocorrelation as a validation step does not allow the detection of periodicities near each others. Nevertheless, in real datasets, most of the periodicities are sufficiently spaced to be detected by the autocorrelation function and thus remains efficient as a validation step.

REAL DATASETS

Benchmarks have also been performed on real datasets (Table 2) and different types of time series have been chosen in order to test the validity of the proposed method.

- **GEF (Hong et al. (2016b)):** This dataset has been provided for the Global Energy Forecasting Competition 2014 (GEFCom2014) Hong et al. (2016a), a probabilistic energy forecasting competition. The dataset is composed of 6 years of hourly load data. This time series is multi-periodic with the following periodicities: daily (24), weekly (168) and bi-annual (4383). The CFD-Autoperiod method has detected and validated 4 periodicities with 3 of them correct. Whereas the autoperiod has detected 5 periodicities with only 2 valid and has missed the long term bi-annual periodicity.

- **Great lakes:** This dataset contains monthly water level of the 5 great lakes and is provided by the National Oceanic and Atmospheric Administration Quinn & Sellinger (1990). This time series is mono-periodic with a periodicity of 12 months. The autoperiod method has

detected 4 different periodicities with only one correct. Among these latter, 24 and 72 periodicities were detected and are only resulting correlations of the 12 periodicity. Whereas the CFD-Autoperiod has successfully filtered out the correlations of the 12 one. Parthasarathy et al. (2006) used this dataset as well but did not write the exact periodicities detected by their method. In their plots, the segmentation for both Ontario and Clair lakes does not correspond to a periodicity of 12.

- **Pseudo periodic (Keogh & Pazzani):** These datasets contain 10 pseudo periodic time series generated from 10 different simulation runs. The data appears to be highly periodic, but never exactly repeats itself. Parthasarathy et al. (2006) did not write their exact results but the segmentation shown on their plot seems to correspond to a detected periodicity of 155. The CFD-Autoperiod method found a periodicity of 144 and the exact true periodicity seems to be 142.

- **Boston Tides:** This dataset contains water level records of Boston, MA from July 01 to August 31, with 6 minutes as sampling interval. It has been recently used by Yuan et al. (2017) to evaluate their method. They successfully detected 2 periodicities but their method required a preprocessing step whereas the CFD-Autoperiod method does not require any. The first detected periodicity is 12,4 hours corresponding to the semi-diurnal constituent of 12 hours and 25.2 minutes. They have also detected 28,5 days and 29 days periodicities which correspond to a lunar month. The CFD-Autoperiod method detected a periodicity of 24 hours and 50 minutes whereas the autoperiod did not detect it. This value is interesting because it corresponds to the behaviour of a mixed tide (when there is a high high tide, a high low tide followed by a low high tide and a low low tide, in 24hour ans 50 minutes). However, it has not detected the lunar month periodicity but this might be due to the lack of data used. Indeed, Yuan et al. (2017) used 2 months of data and the CFD-Autoperiod can only detect periodicities of a length inferior or equal to the half of the signal length.

| | | Detected periodicities | |
| --- | --- | --- | --- |
| | | Values | Nb |
| GEF | Fourier transform | 5.99, 11.99,... | 44 |
| | autoperiod | 24, 168, 192, 288, 528 | 5 |
| | CFD-Autoperiod | 24, 171, 496, 4118 | 4 |
| Great lakes Clair | Fourier transform | 12.19, 23.27, 28.44,... | 7 |
| | autoperiod | 12, 24, 35, 72 | 4 |
| | CFD-Autoperiod | 12, 505 | 2 |
| Great lakes Ontohoio | Fourier transform | 12.2, 28.4, 36.6, ... | 6 |
| | autoperiod | 12, 35, 72 | 3 |
| | CFD-Autoperiod | 12 | 1 |
| pseudo 1 | Fourier transform | 18.7, 35.1, 74.1, 142.9, ... | 4 |
| | autoperiod | 74, 146 | 2 |
| | CFD-Autoperiod | 144 | 1 |
| Boston Tide | Fourier transform | 120.0, 120.9, 121.9, ... | 11 |
| | autoperiod | 124 | 1 |
| | CFD-Autoperiod | 125, 246 | 2 |

Table 2: Detected periodicities on real Dataset

## CONCLUSION AND FUTURE WORK

This paper describes an algorithm called CFD-Autoperiod detecting periodicities in time series and improving the autoperiod method proposed in Vlachos et al. (2005). CFD-Autoperiod can be applied on noisy time series containing multiple periodicities and output raw periodicities that can later be refined by external domain specific knowledge (for instance 24h for human daily activities). One case not treated in this study concerns non-stationary series. A possible technique would consists in tracking the evolution of the periodicities through time and using a Kalman filter to track the apparition, disappearance or evolution of the detected periodicities. Using the confidence of the Kalman filter we could decide whether to continue considering the presence of a particular periodicity in

the signal even if it is not detected for a while. This would strengthen the results obtained by CFD-Autoperiod and give more reliable periodicities. Thus, even more complex machine learning models can be built on top of them.

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
