# OpenReview forum: "A fully automated periodicity detection in time series"
_ICLR.cc/2019/Conference_

### Official Review · AnonReviewer1 · 2018-11-02
**An ad-hoc method for period detection. Possibly unfit for this conference**

**Rating:** 3
**Confidence:** 2

**Review:**

This paper introduces a method to do period detection. It builds off of the autoperiod method by adding density clustering, a lowpass filter, and a linear detrending after auto correlation.

The results section was very vague. The process of  generating the synthetic signals was not specific. There were no visualizations, which would help the reader understand how this method performs better. Visualizations would have been especially useful for the real datasets.

Having said this, I don't think this paper is fit for this conference.

---

### Official Review · AnonReviewer3 · 2018-11-11
**This paper would not be best fit to ICLR.**

**Rating:** 5
**Confidence:** 2

**Review:**

This paper focuses on the extraction of the (multi) periodicities from a signal. The paper describes the conventional method based on the Fourier transformation and/or autocorrelation methods, and proposed method, which first detects a distribution of spectral leakages, and prune the periodicity hints by using a clustering algorithm. The proposed method is also extended to deal with multi-periodicities. The effectiveness of the proposed method is shown with the controlled simulation data and several real data. This paper is well written (note it is over 8 pages though), but it is not learning-based approach, and would not best fit to major ICLR interests.

Comments:
- The abstract needs to be more self-consistent without referring the citation for a brief explanation. Also it should have more detailed experimental discussions.
- Algorithm 1 needs some refinement (too code-like, although it is understandable). For example, several methods (nextBinValue and append) would be better to be replaced with other (human readable) expressions.

---

### Official Review · AnonReviewer2 · 2018-11-15
**A heuristic method for time series periodicity detection; topic not matching scope of ICLR**

**Rating:** 3
**Confidence:** 3

**Review:**

The authors present a heuristic method to detect periodicity in time series. It extends a previous approach in dealing with noise and the setting of multiple periodicities.

The topic does not match the scope of ICLR and would be better suited for a different venue.

The method is demonstrated in a purely experimental fashion. However, without detailed inspection of the datasets it remains unclear in what cases the heuristics apply and where they fail. A more thorough analysis of the robustness of the algorithm is necessary, in particular a detailed presentation of failure cases.

---

### Meta-Review · Area_Chair1 · 2018-12-17
**not fit for ICLR**

**Confidence:** 5
**Recommendation:** Reject

**Metareview:**

This paper presents an heuristic method to detect periodicity in a time-series such that it can handle noise and multiple periods.

All reviewers agreed that this paper falls off the scope of ICLR since it does not discuss any learning-related question. Moreover, the authors did not provide any response nor updated manuscript addressing the reviewers remarks. The AC thus recommends rejection.